# Serotonin modulates infraslow oscillation in the dentate gyrus during non-REM sleep

Gergely F Turi[1,2]*[†], Sasa Teng[3,4][†], Xinyue Chen[4,5], Emily CY Lim[6], Carla Dias[1,2], Ruining Hu[3], Ruizhi Wang[4], Fenghua Zhen[3][‡], Yueqing Peng[3,4,7]*

[1]New York State Psychiatric Institute, Division of Systems Neuroscience New York, New York, United States; [2]Department of Psychiatry, Vagelos College of Physicians and Surgeons, Columbia University, New York, United States; [3]Institute for Genomic Medicine, Vagelos College of Physicians and Surgeons, Columbia University, New York, United States; [4]Department of Neurology, Vagelos College of Physicians and Surgeons, Columbia University, New York, United States; [5]Department of Neuroscience, Vagelos College of Physicians and Surgeons, Columbia University, New York, United States; [6]Columbia College, Columbia University, New York, United States; [7]Department of Pathology and Cell Biology, Vagelos College of Physicians and Surgeons, Columbia University, New York, United States

*For correspondence:
gt2253@cumc.columbia.edu (GFT);
yp2249@cumc.columbia.edu (YP)

[†]These authors contributed equally to this work

Present address: [‡]National Institute of Allergy and Infectious Diseases, Bethesda, United States

Competing interest: The authors declare that no competing interests exist.

## eLife Assessment

This **important** study shows that a very slow (infraslow) oscillation occurs in calcium recordings from the dentate gyrus of the adult mouse. The authors suggest that it is related to sleep stage and serotonin acting at one type of serotonin receptor in the dentate gyrus. The results are significant because they suggest new insight into how a slow oscillation affects memory through serotonin receptors in the dentate gyrus. **Convincing** data are provided to support the claims.

**Abstract** Synchronous neuronal activity is organized into neuronal oscillations with various frequency and time domains across different brain areas and brain states. For example, hippocampal theta, gamma, and sharp wave oscillations are critical for memory formation and communication between hippocampal subareas and the cortex. In this study, we investigated the neuronal activity of the dentate gyrus (DG) with optical imaging tools during sleep-wake cycles in mice. We found that the activity of major glutamatergic cell populations in the DG is organized into infraslow oscillations (0.01–0.03 Hz) during NREM sleep. Although the DG is considered a sparsely active network during wakefulness, we found that 50% of granule cells and about 25% of mossy cells exhibit increased activity during NREM sleep, compared to that during wakefulness. Further experiments revealed that the infraslow oscillation in the DG was correlated with rhythmic serotonin release during sleep, which oscillates at the same frequency but in an opposite phase. Genetic manipulation of 5-HT receptors revealed that this neuromodulatory regulation is mediated by *Htr1a* receptors and the knockdown of these receptors leads to memory impairment. Together, our results provide novel mechanistic insights into how the 5-HT system can influence hippocampal activity patterns during sleep.

## Introduction

Sleep is an evolutionarily conserved biological process observed in the animal kingdom. Invertebrates such as *Drosophila*, *C. elegans*, and even *Hydra* show sleep-like behavior (*Cirelli et al., 2005*; *Koh et al., 2008*; *Raizen et al., 2008*; *Guo et al., 2016*; *Kanaya et al., 2020*). In mammals, sleep stages can be subdivided into rapid eye movement (REM) sleep and non-REM (NREM) phases. During NREM sleep the activity of the skeletal muscle is reduced and the EEG is dominated by slow, high amplitude synchronous oscillations while in REM sleep the EEG activity is more similar to the awake brain state. There are also smaller building blocks of sleep stages. For instance, NREM sleep is often interrupted by brief epochs, which based on their EEG and EMG signatures, show remarkable reminiscence to awake states, thus, they are often referred to as microarousals (MAs). These events can be captured as brief motor bursts on the EMG signal and by abrupt alternations of low frequency oscillatory patterns in EEG recordings (*Lo et al., 2004*; *Bartsch et al., 2012*; *dos Santos Lima et al., 2019*). Even though MAs are natural parts of the sleep architecture, the function of these events is largely unknown.

Sleep is a fundamental biological process that plays a crucial role in many physiological functions. Studies using various model organisms have demonstrated that severe sleep loss or total sleep deprivation can even have fatal effects (*Bentivoglio and Grassi-Zucconi, 1997*; *Rechtschaffen et al., 1983*; *Shaw et al., 2002*). While the exact biological function of sleep is not fully understood, research has suggested that it plays a crucial role in memory consolidation, which is the process of converting newly acquired memories into a permanent form. The supportive evidence for this hypothesis is quite rich although far from consolidated. Current models suggest that slow wave sleep, which makes up deeper stages of NREM sleep, processes declarative memory and it requires the active participation of the hippocampus through the replay of episodic memory traces captured during waking hours (*Buzsáki, 1989*; *Klinzing et al., 2019*). While the role of hippocampal CA1 and CA3 subareas in this process is extensively studied, less is known about the contribution of the dentate gyrus (DG) which is the first station of the classic trisynaptic loop and located upstream from CA3. The main glutamatergic cell types of the DG are the granule cells (GCs) located in the granule cell layer and the mossy cells (MCs) which populate the hilus or polymorph layer between the upper and lower blades of the GC layers (*Amaral, 1978*). Experimental work has shown that GCs and MCs form a functional unit to perform pattern separation during active exploration (*Senzai and Buzsáki, 2017*; *GoodSmith et al., 2017*; *Danielson et al., 2017*), a neuronal mechanism by which distinct memory traces can be created even if the input pattern is highly overlapping. Furthermore, it has been also shown that in awake immobile mice, the DG generates sparse, synchronized activity patterns driven by inputs from the entorhinal cortex (*Pofahl et al., 2021*).

Sleep/wake cycles are regulated by various monoaminergic and peptidergic cell groups located throughout the hypothalamus and brainstem. Among the monoaminergic neuromodulators, the serotonin (5-HT) system has been shown to promote wakefulness and suppress REM sleep (*Monti, 2011*; *Jouvet, 1999*). It is widely accepted in the field that 5-HT release is reduced during NREM sleep, and 5-HT cells become entirely silent during REM sleep (*Jouvet, 1999*; *Lee and Dan, 2012*). Serotonin is also recognized as one of the primary neuromodulatory inputs to the DG and a substantial body of evidence implicates that genetic manipulation of 5-HT or 5-HT-related genes has a profound impact on a broad range of hippocampal processes, including anxiety behavior, learning and memory (*Luchetti et al., 2020*; *Yoshida et al., 2019*; *Santarelli et al., 2003*; *Caspi et al., 2010*; *Holmes et al., 2003*; *Teixeira et al., 2018*). Serotonin receptors are expressed throughout the dorsoventral axis of the hippocampus (*Tanaka et al., 2012*), with GCs expressing both the inhibitory *Htr1a* and excitatory *Htr4* receptor subtypes and about a quarter of MCs expressing the excitatory *Htr2a* receptors (*Tanaka et al., 2012*). Furthermore, the mesopontine Median Raphe Nucleus plays a role in memory-related hippocampal ripple oscillations (*Wang et al., 2015*; *Jelitai et al., 2021*). Together these findings suggest that the 5-HTergic system has a strong impact on hippocampal functions.

In this study, we investigated the activity of GCs and MCs during sleep-wake cycles and their modulation by the 5-HT system. We found that both glutamatergic cell types – especially GCs – exhibit higher activity during specific sleep stages compared to wakefulness, and the calcium activity is organized to infraslow oscillatory (ISO) cycles during NREM sleep epochs. Our further findings obtained by recording from major 5-HTergic cell populations in the raphe and by manipulating specific 5-HT receptor subtypes suggest that the slow oscillatory activity in the DG is regulated by an ISO of the 5-HT system oscillating in the opposite phase during NREM which is also tightly coupled to

MA events. Genetic knockdown of *Htr1a* receptors in the DG impairs the ISOs and contextual fear memory, suggesting a key role for this inhibitory 5-HT receptor in governing the oscillatory effect.

## Results
### Infraslow neural oscillation of DG populations during NREM sleep

To investigate population activity during sleep and awake brain states in the DG in a cell type-specific manner, we injected a set of *Dock10*[Cre+/-] mice with AAV1-FLEX-GcaMP6s to drive the expression of GCaMP specifically in GCs (*Kohara et al., 2014*). The mice were then implanted with fiber photometry probes, EEG, and EMG electrodes to facilitate brain state classification. Two weeks after recovery, we conducted chronic photometry and EEG recordings while the animals experienced natural wake/ sleep cycles in a behavioral chamber. Consistent with previous studies (*Senzai and Buzsáki, 2017*; *Pofahl et al., 2021*; *Shen et al., 1998*), we observed significantly higher populational calcium activity during sleep states (NREM and REM), compared to wakefulness (*Figure 1A–C*). Strikingly, our data also revealed a structured pattern in the calcium signal: GC activity was organized to an infraslow oscillation (ISO, 1–2 cycles/min, or 0.017–0.033 Hz) during NREM sleep (*Figure 1A and D*). The oscillation amplitude and power remained largely unchanged in the early and late stages of NREM sleep epochs. However, the analysis of the oscillatory power and amplitude in the early stage of the first NREM sleep epochs following prolonged wakefulness yielded statistically significant differences (*Figure 1—figure supplement 1A, B*). Cross-correlation analysis between calcium activity and EMG/ EEG revealed a strong correlation between the calcium oscillation and the sigma band of the EEG (*Figure 1—figure supplement 1A, B*). Additionally, the troughs of the calcium ISO partially coincided with MA episodes (*Figure 1E*). Quantitative analysis showed that 29% of the ISO events during NREM sleep were followed by a MA epoch while 62% of ISO events were accompanied by the maintenance of NREM sleep (*Figure 1F*). Notably, the reduction in GC activity significantly preceded the occurrence of MA events as judged from the EMG, where the average latency between the calcium trough and the onset of the MA was 6.06+/-0.31 s (*Figure 1G and H*).

MCs are among the first synaptic partners of GCs, therefore, we set out to record the calcium activity of MCs by injecting a new cohort of mice with dopamine 2 receptor-Cre (*Drd2*[Cre+/-]) genetic background (*Figure 2A and B*). *Drd2* expression in the hilus is highly specific to MCs (*Puighermanal et al., 2015*; *Gangarossa et al., 2012*). MCs displayed similar levels of calcium activity during wake and NREM sleep, and significantly increased activity during REM sleep (*Figure 2C*). Similar to GCs, we observed the ISO in MCs calcium signal, although with a slightly lower peak frequency (*Figure 2A and D*). Furthermore, correlation analysis between photometric and EEG/EMG signals showed similar results as with GCs, i.e., around 30% of ISO events in MCs lead to MAs and 60% resulted in the maintenance of NREM (*Figure 2E–G*). The average latency between the onset of calcium declines and the onset of MAs was somewhat shorter than in GCs (5.18+/-0.49 s) (*Figure 1H*).

Overall, our data showed similar NREM-specific oscillatory activity in both GCs and MCs. The increased activity of GCs and MCs during sleep stages is consistent with previous studies (*Senzai and Buzsáki, 2017*; *Pofahl et al., 2021*; *Shen et al., 1998*; *Jung and McNaughton, 1993*). However, the ISO in the calcium activity of glutamatergic neurons, to the best of our knowledge, has not been reported yet.

### Ensemble activity of dentate gyrus in sleep

To study the cellular mechanisms underlying sleep-specific activity, we conducted head-fixed two-photon imaging experiments combined with EEG/EMG recordings to record neuronal ensemble activity in the DG during sleep stages (*Figure 3A*). Wild-type mice were injected with AAVdj-SYN-jGCaMP7b in the DG followed by optical window, EEG, and EMG implantations. After a few weeks of recovery, we recorded the cellular activity in the granule cell layer and in the hilus (*Figure 3A–C*) while the mouse was engaged in locomotory and motionless epochs on a treadmill. To facilitate sleep under the two-photon microscope, the mice underwent mild sleep deprivation before the recordings (see Methods for details). *Post hoc* analysis of EEG/EMG signals confirmed the presence of sleep states (*Figure 3B and C*, also *Figure 3—figure supplement 1A, B*). Comparing the calcium signals recorded during awake and sleep intervals revealed that the calcium activity was significantly upregulated during NREM in about 50% of the GC and 28% of the putative MC populations (*Figure 3D and*

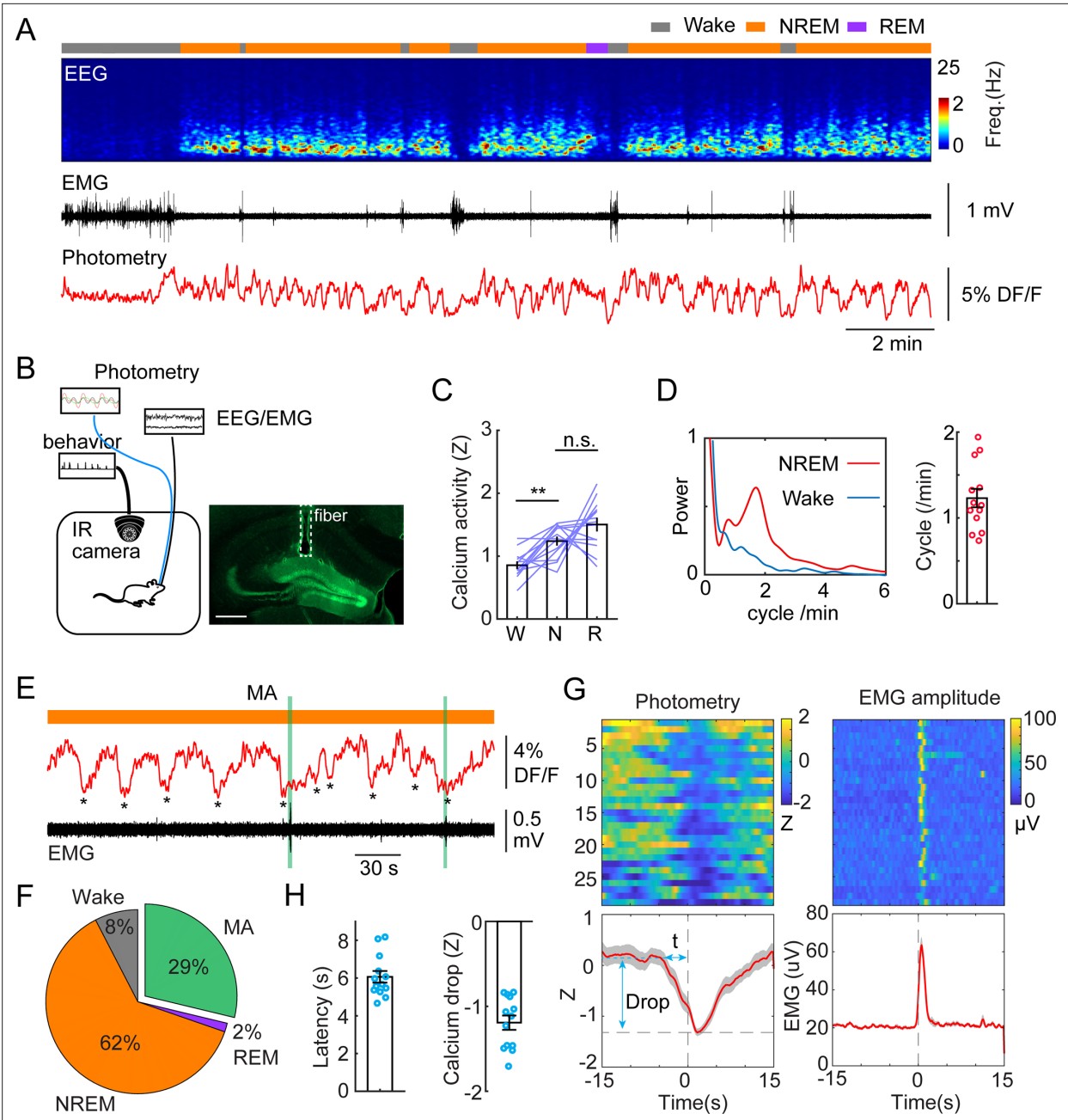

**Figure 1.** Infraslow calcium oscillation of granule cells during non-rapid eye movement (NREM) sleep. (**A**) Representative recording session showing infraslow oscillation (ISO) during NREM sleep. From top to bottom: brain states, EEG power spectrogram (0–25 Hz), EMG amplitude, photometric signal. (**B**) Left, Schematic representation of the recording setup. Right, a fluorescence image showing the expression of GCaMP6s (green) in the granule cell layer and the optic fiber placement (dashed line) in a *Dock10*^Cre mouse. Scale bar, 500 µm. (**C**) Quantification of calcium activity in granule cells (13 recording sessions in 5 *Dock10*^Cre mice, n.s. – no significance, **p<0.01, paired t-test) in wake (W), NREM sleep (N), and rapid eye movement (REM) sleep (R). (**D**) Left: oscillation peak frequency during NREM sleep and wake based on Fourier transformation of the photometry signal. Right: Quantification of calcium oscillations in granule cells (GCs) (13 sessions in 5 mice). (**E**) A representative example showing the coincidence of calcium troughs (indicated by *) with microarousals (MAs). (**F**) Percentage of state transition outcome from each calcium dip. (**G**) Peri-stimulus time histogram (PSTH) in one recording session showing calcium signal aligned with the onset of MAs. Bottom left: parameters of the calcium signal used for quantification. (**H**) Quantification of the latency (t) and magnitude of the calcium trough (Drop) during MAs (13 sessions from 5 *Dock10*^Cre mice).

The online version of this article includes the following figure supplement(s) for figure 1:

**Figure supplement 1.** Characterization of infraslow oscillation of granule cells (GC) during non-rapid eye movement (NREM) sleep.

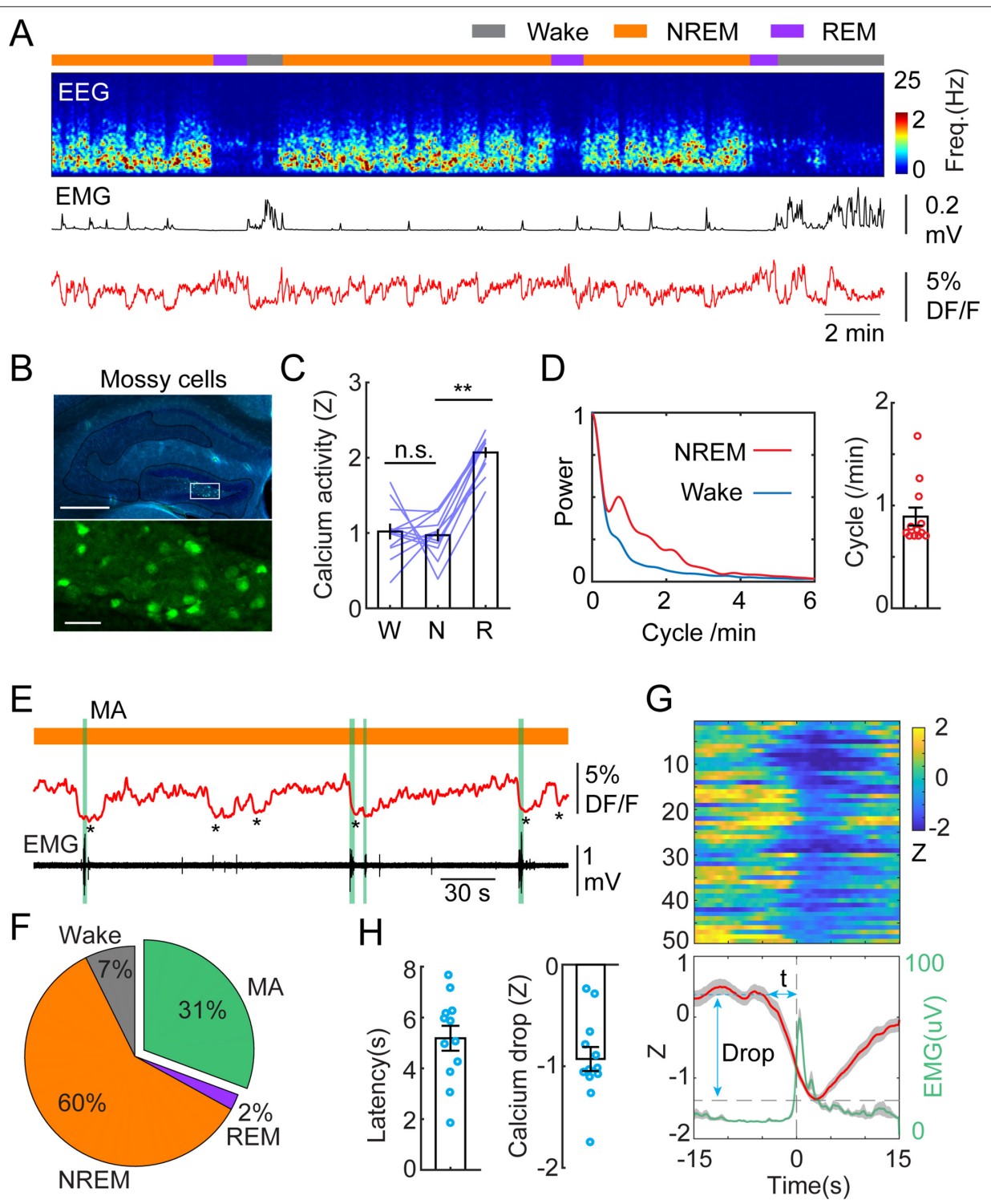

**Figure 2.** Infraslow calcium oscillation of mossy cells during non-rapid eye movement (NREM) sleep. (**A**) Representative recording session showing infraslow calcium oscillation during NREM sleep in a *Drd2*^Cre+/- mouse injected with AAV-FLEX-GCaMP6s. From top to bottom: brain states, EEG power spectrogram (0–25 Hz), EMG amplitude, photometric signal. (**B**) Fluorescence images showing the expression of GCaMP6s (green) in the mossy cells in the area marked with the white rectangle. Blue, DAPI. Scale bars, 500 μm and 50 μm. (**C**) Quantification of calcium activity in mossy cells during different brain states (12 recording sessions in 4 *Drd2*^Cre+/- mice; n.s. – no significance, **p<0.01, paired t-test). (**D**) Left: oscillation peak frequency during NREM sleep based on Fourier transformation of the photometry signal. Right: quantification of calcium oscillations during NREM sleep. (**E**) A representative example showing the coincidence of calcium troughs (indicated by *) with microarousals (MAs) (represented with vertical green lines). (**F**) Percentage of

*Figure 2 continued on next page*

*Figure 2 continued*

state transition outcome from each calcium dip. (**G**) Peri-stimulus time histogram (PSTH) from one recording session showing calcium signals (red) and EMG signals (light green) aligned with the onset of MAs. (**H**) Quantification of the latency (t) and the magnitude of calcium troughs (Drop) during the MAs (12 sessions from 4 *Drd2*^Cre mice).

---

*E*) in the hilus. This finding is consistent with a previous two-photo imaging study showing that GCs display high activity during running but also during resting periods (*Pilz et al., 2016*). Similar to the photometry experiments, we then correlated the MAs with single-cell calcium activity. We found that up-regulated, but not down-regulated putative GCs and putative MCs displayed decreased activity during MAs in NREM sleep (*Figure 3D and E*, *Figure 3—figure supplement 1C–E*).

## ISO is driven by serotonin release during NREM sleep

The prominent electrographic patterns in the DG during NREM sleep include dentate spikes and sharp-waves ripples (*Bragin et al., 1995*; *Buzsáki, 1986*; *Farrell et al., 2024*), but these events are generally too short-lived (30–120 ms) to account for the long-lasting active periods during ISO. Given the second-long latencies between calcium troughs and MAs, we turned our attention toward neuro-modulatory systems that are thought to modulate sleep/wake periods on a slower scale and display phasic activity during sleep cycles (*Scammell et al., 2017*). An earlier study in mice identified phasic activity of serotonergic neurons in the dorsal raphe during NREM sleep (*Oikonomou et al., 2019*). This finding was corroborated by another group, which also demonstrated that the frequency range of the phasic activity falls in the infraslow regime (0.01–0.3 Hz) (*Kato et al., 2022*). Therefore, we looked at the possibility whether the 5-HT system might be responsible for the modulation of the ISO in the DG. In a subsequent set of experiments, we used GRAB$_{5-HT2h}$, a recently developed and optimized genetically encoded 5-HT sensor (*Wan et al., 2021*) to measure 5-HT dynamics during sleep. Considering that 5-HT is exclusively synthetized by the raphe nuclei in the adult brain (*Muzerelle et al., 2016*), our initial step involved stereotactic injection of an AAV expressing the GRAB$_{5-HT2h}$ sensor into this region, followed by the implantation of an optic fiber above the injection site. Our results recapitulated the canonical view on 5-HT dynamics during sleep/wake cycles: we observed the highest 5-HT level during wakefulness, significantly lower level during NREM and detected the lowest 5-HT levels during REM periods (*Figure 4—figure supplement 1A, F*; *Monti, 2011*; *Jouvet, 1999*; *Lee and Dan, 2012*). We also observed a phasic pattern of 5-HT during NREM sleep (*Figure 4—figure supplement 1A, E*) which is consistent with recently published studies (*Kato et al., 2022*; *Wan et al., 2021*). Our further analyses revealed that the rhythmic 5-HT release during NREM sleep was also correlated with MA episodes (*Figure 4—figure supplement 1B*). Correlation analysis revealed that 35% of phasic 5-HT peaks during NREM sleep were accompanied by MA events detected by the EMG bursts (*Figure 4—figure supplement 1C*). The latency between the phasic increase of 5-HT and the MA onset was 6.82+/-0.44 s (*Figure 4—figure supplement 1D, E*).

To confirm that this infraslow oscillatory pattern in 5-HT is manifested in the hippocampus as well, we injected AAV-GRAB$_{5-HT2h}$ into the DG and implanted an optic fiber above the injection site. (*Figure 4A*). Similar to 5-HT signals recorded from the raphe nuclei, we observed the highest level of 5-HT in wake, the lowest in REM sleep, and intermediate level in NREM sleep (*Figure 4B–D*). Importantly, the 5-HT dynamics during NREM sleep displayed the same oscillatory pattern in the DG as that we observed in the raphe (*Figure 4B*). Thirty-four percent of the phasic increases in 5-HT were associated with MAs (*Figure 4C*). Furthermore, the frequency of phasic 5-HT release during NREM was 1.14+/-0.04 cycle/min (*Figure 4E*), closely matching the frequency of the ISO in the DG.

The tight alignment of both the calcium troughs in the DG and the phasic 5-HT peaks in the raphe with MA events suggest a causal relationship between them. To test this hypothesis, we conducted dual site fiber photometry recordings in *Slc6a4*^Cre+/- mice by injecting AAV1-FLEX-GCaMP6s in the raphe and with AAV9-CaMKII-GCaMP6s in the DG (*Figure 5A*) followed by fiber implants to these areas. *Post hoc* histological analysis of AAV9-CaMKII-GCaMP6s injected brains confirmed that the expression of GCaMP is largely restricted to GCs (*Figure 5—figure supplement 1*). Furthermore, analysis of the calcium signals from the AAV9-CaMKII-GCaMP6s-labeled population revealed a similar ISO pattern as we observed with the genetically more selective approach (*Figure 5—figure supplement 1*). Strikingly, the phases of the photometry signals in the DG and raphe appeared to be anti-correlated during NREM sleep, i.e., the calcium peaks in the raphe appeared when the intensity

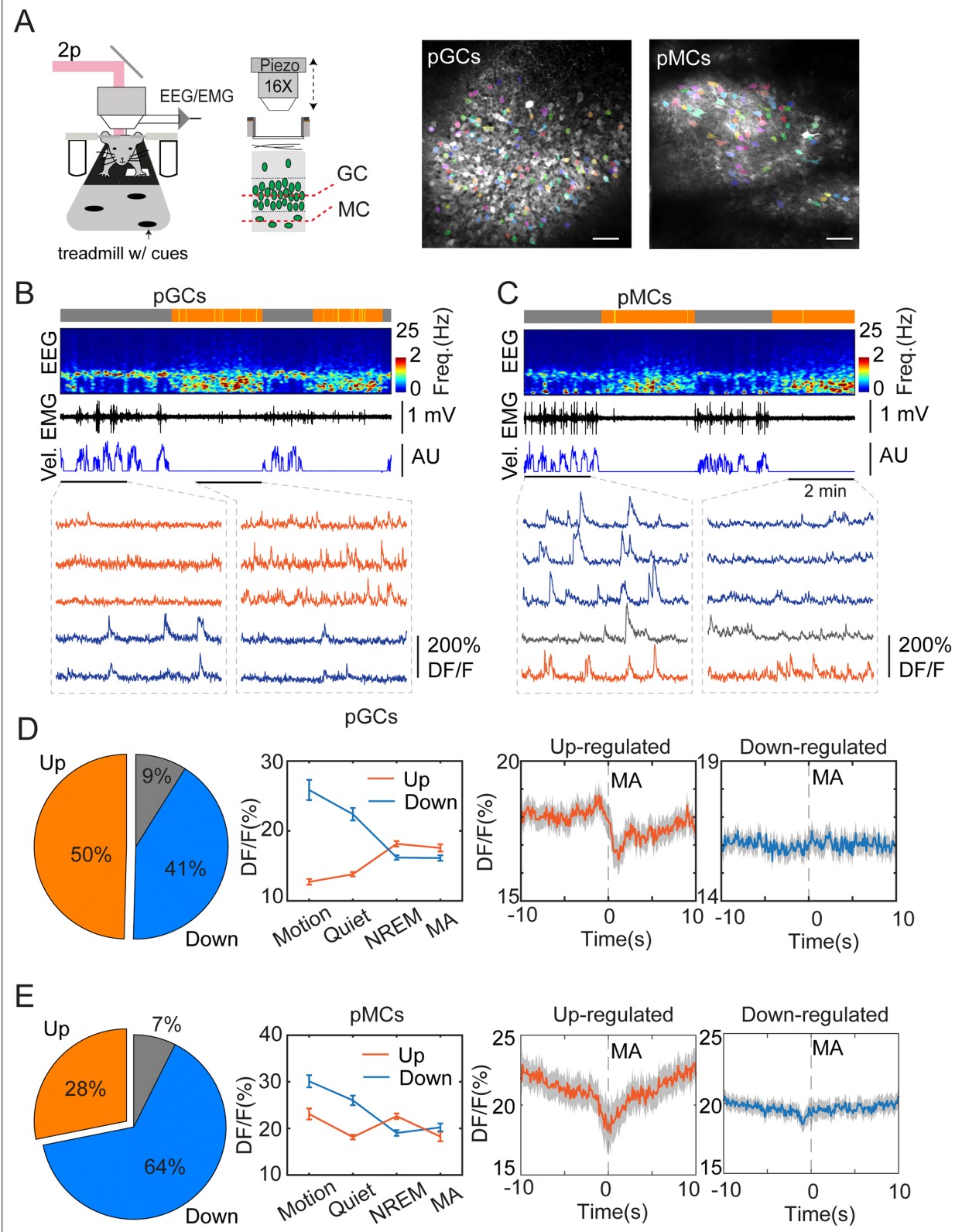

**Figure 3.** Two-photon calcium imaging of dentate gyrus (DG) activity in sleeping mice. (**A**) Left, schematic layout of two-photon imaging and EEG recording setup. Right, representative images showing putative granule cells (pGCs) and putative mossy cells (pMCs). Scale bars, 50 μm. (**B, C**) Representative two-photon recording sessions of field of views containing putative pGCs pMCs. From top to bottom: brain states (gray – awake, orange – NREM), EEG spectrogram (0–25 Hz), EMG, velocity (vel.), and calcium traces in individual cells. Red traces - upregulated cells, blue traces –

*Figure 3 continued on next page*

*Figure 3 continued*

downregulated cells, gray traces – non-significant cells. (**D, E**) Left, Percentage of Up-, downregulated, and non-significant cells from the entire recorded cell populations. Middle, quantification of calcium activity in different brain states in up-, down-regulated cells (pGCs: p<0.05 in up-regulated cells between non-rapid eye movement (NREM) and microarousal (MA), p=0.75 in down-regulated cells between NREM and MA; pMCs: p<0.001 in up-regulated cells between NREM and MA, p=0.16 in down-regulated cells between NREM and MA, paired t-test). Right, Averaged activity in up-, down-regulated pGCs and pMCs during microarousals (MA). Time 0 indicates the MA onset. Putative GCs: 369 cells from 3 C57BL/6 J mice; putative MCs: 135 cells from 2 C57BL/6 J mice.

The online version of this article includes the following figure supplement(s) for figure 3:

**Figure supplement 1.** Two-photon calcium imaging in sleeping mice.

of the calcium signal dropped in the DG (*Figure 5B*). Indeed, Pearson's correlation analysis of the two signals resulted in a strong negative correlation between the raphe and DG during NREM sleep (*Figure 5C and D*). Interestingly, during wake periods, the DG activity was positively correlated with the raphe activity (*Figure 5D*). Contrary to NREM sleep, the correlated activity between the dentate gyrus and raphe displayed a large variability during REM sleep. Together, these results suggest that the descending phase of the population activity during the ISO is driven by the phasic release of 5-HT during NREM sleep.

## Granule cell activity is inhibited via *Htr1a* receptors during the ISO

Our results so far suggested that the periodic increase in 5-HT levels during NREM sleep inhibits GC and MC activity. To identify the molecular basis of this relationship, we conducted local genetic manipulations of the *Htr1a* receptors. We selected this subtype because of its high expression pattern in GCs (*Tanaka et al., 2012*) and its well-characterized inhibitory effect on this cell type (*Tsetsenis et al., 2007*). We injected a cohort of *Htr1a*<sup>flox/flox</sup> mice (*Samuels et al., 2015*) with a mix of AAV1-hSyn-Cre and AAV9-CaMKII-GCaMP6s to simultaneously express a calcium sensor and to downregulate *Htr1a* receptors specifically in the GCs (*Figure 6A*). A control group of *Htr1a*<sup>flox/flox</sup> mice was injected with AAV9-CaMKII-GcaMP6s. We then conducted fiber photometry recordings in the DG. Analysis of the calcium signals during awake and sleep period resulted in significantly decreased ISO during NREM in the *Htr1a* downregulate group compared to the control (*Figure 6B–D*) Post hoc in situ hybridization confirmed the lack of *Htr1a* receptors at the injection site (*Figure 6—figure supplement 1*). To examine if *Htr1a* knock-down affects sleep architecture, we quantified the total duration of wake, NREM sleep, REM sleep, and frequency of MAs. No significant difference was observed between *Htr1a*<sup>flox/flox</sup> and control mice (*Figure 6—figure supplement 2A*). We reasoned that Htr1a receptors are required for the inhibitory response of DG activity during the MAs. Indeed, we found that *Htr1a*<sup>flox/flox</sup> mice displayed lower calcium drops and longer latency during the MAs, compared to that in control mice (*Figure 6—figure supplement 2B, C*).

Together, our two-site recordings and genetic manipulation data indicated that the ISO in the DG is modulated through an inhibitory mechanism via *Htr1a* receptors.

## 5-HTergic modulation of the DG via *Htr1a* receptors is required for contextual memory

Neural oscillations during NREM sleep are thought to be involved in memory consolidation (*Girardeau and Lopes-Dos-Santos, 2021*). We hypothesized that the *Htr1a*-dependent ISO is a required component for hippocampus-dependent memory consolidation. To test this hypothesis, we genetically knocked out *Htr1a* receptors in the DG and examined its effect on memory performance. We injected *Htr1a*<sup>flox/flox</sup> mice with AAV9-CaMKII-Cre-GFP (intervention group) or with AAV9-CaMKII-GFP (control group) bilaterally in the dorsal DG. Two weeks after surgery, we tested the memory performance of the mice by examining freezing behavior in a contextual fear conditioning (CFC) paradigm (*Figure 6E*). In accordance with our hypothesis, we observed diminished memory performance in retrieval tests in the intervention group (*Figure 6F*) measured by the time spent with freezing (30.0 ± 4.63% *vs* 49.2 ± 7.10%).

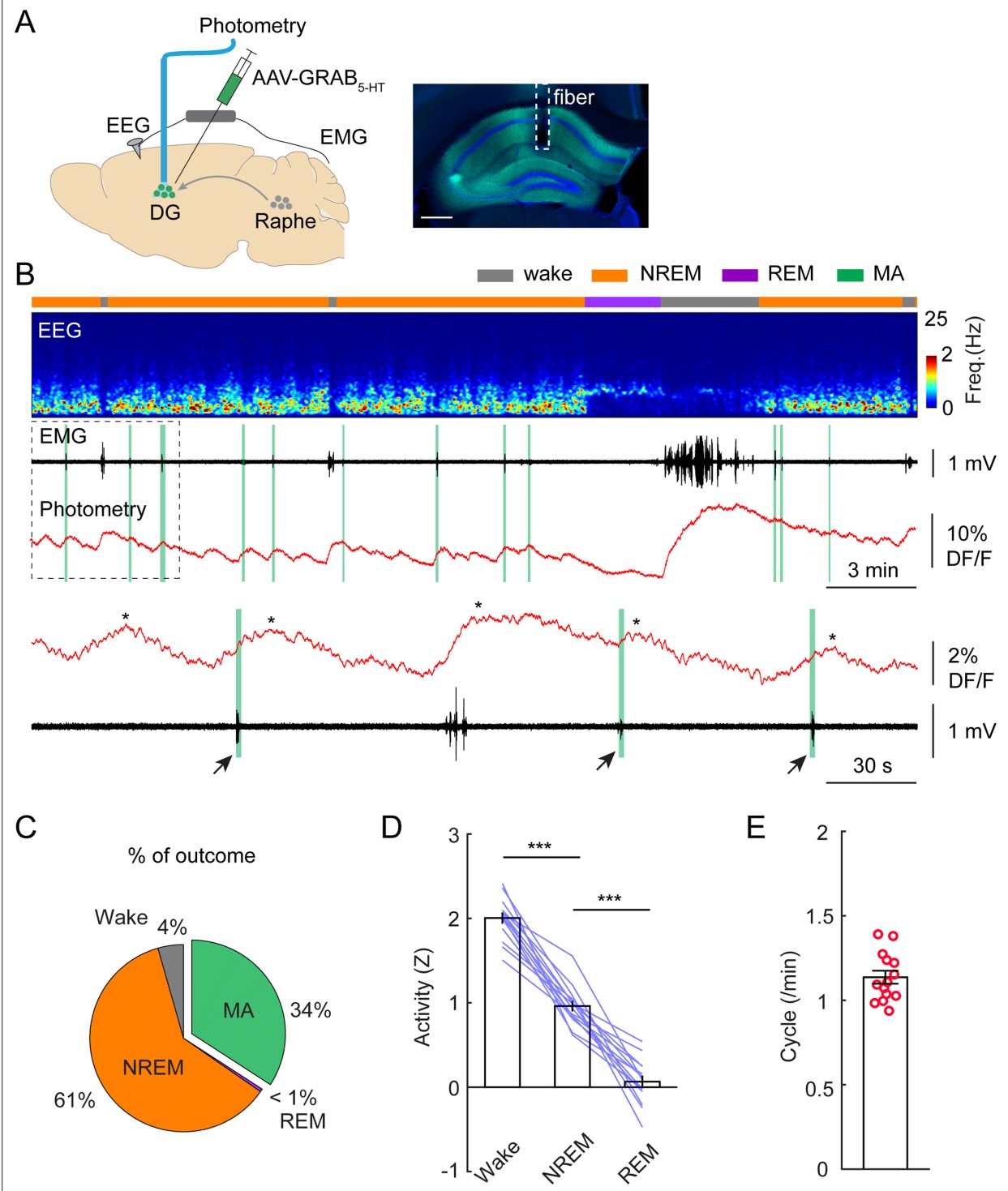

**Figure 4.** Phasic release of 5-HT in the dentate gyrus (DG) during non-rapid eye movement (NREM) sleep. (**A**) Left, schematic of experimental design. Right, expression of 5-HT sensor in the hippocampus. Scale bar, 500 μm. (**B**) A representative example of 5-HT signals during different brain states. From top to bottom: brain states, EEG power spectrogram (0–25 Hz), EMG signal, photometric signal. The dashed box is enlarged below in panel B. The asterisks indicate the peaks of 5-HT signals. Note the coincidence of 5-HT release with MAs during NREM sleep (black arrows and vertical green lines). (**C**) Percentage of state transition outcome from each 5-HT event (averaged data from 6 mice). (**D**) Quantification of 5-HT signals in the DG during different brain states (14 sessions from 6 C57BL/6 J mice, **p<0.01, ***p<0.001, paired t-test). Data were normalized to Z scores in each recording session. (**E**) Quantification of oscillatory cycles of 5-HT signals in the DG (14 recording sessions from 6 C57BL/6 J mice).

The online version of this article includes the following figure supplement(s) for figure 4:

**Figure supplement 1.** Phasic release of 5-HT in the raphe nuclei during non-rapid eye movement (NREM) sleep.

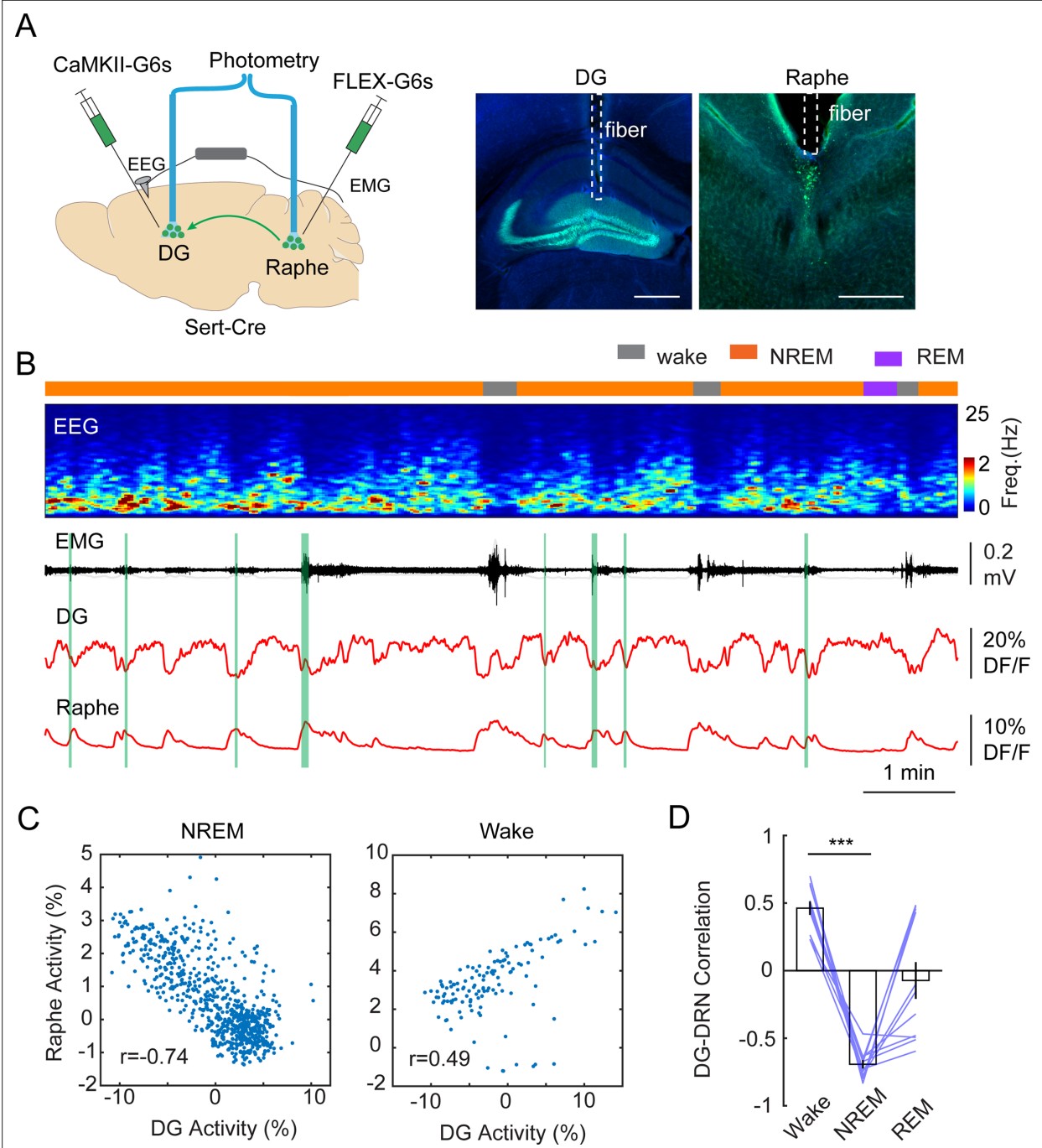

**Figure 5.** Correlation between DG oscillation and activity of 5-HT neurons during non-rapid eye movement (NREM) sleep. (**A**) Left, Schematic representation of the two-site photometry experimental design. Right, Expression of CaMKII-GCaMP6s and fiber placement in the dentate gyrus (DG) and raphe nuclei. Scale bars, 500 μm. (**B**) A representative example of concurrent recording of DG and raphe 5-HT neurons in a *Slc6a4*[Cre+/-] mouse during sleep. From top to bottom: brain states, EEG power spectrogram (0–25 Hz), EMG amplitude, photometric calcium signals (CaMKII-G6s) in DG and in dorsal raphe. (**C**) Correlation analysis of calcium activity between DG and raphe 5-HT neurons during NREM sleep and wakefulness in one recording session. (**D**) Quantification of correlation coefficient between DG activity and raphe activity during different brain states (11 sessions from 3 *Slc6a4*[Cre+/-] mice, ***p<0.001, paired t-test).

The online version of this article includes the following figure supplement(s) for figure 5:

**Figure supplement 1.** CaMKII-labeled cells in the dentate gyrus (DG) display oscillatory activity during non-rapid eye movement (NREM) sleep.

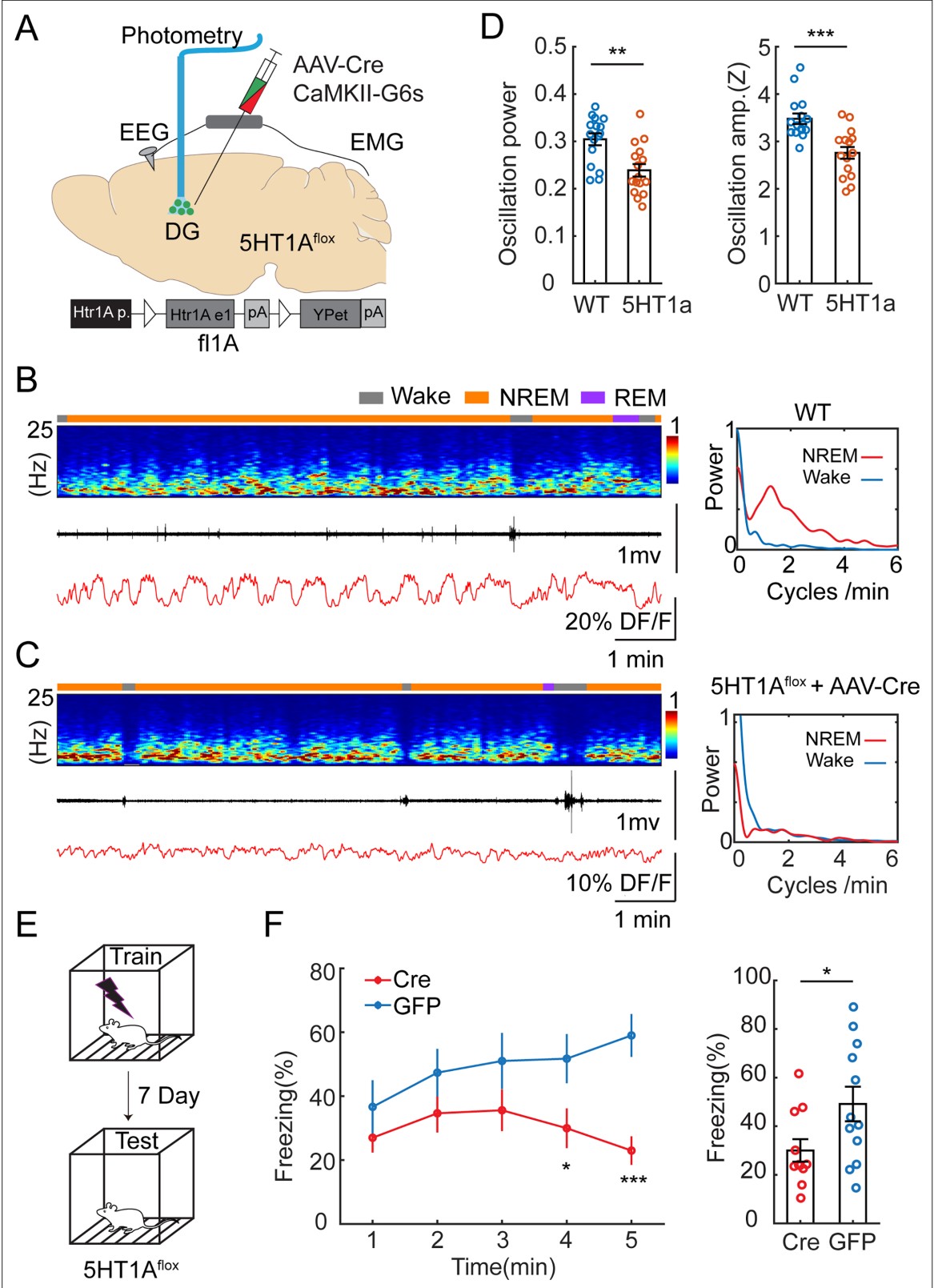

**Figure 6.** Genetic knockdown of *Htr1a* receptors in dentate gyrus (DG) impairs infraslow oscillatory (ISO) and memory performance. (**A**) Schematic representation of the experimental design. A mix of AAV9-CamKII-GCaMP6s and AAV1-hSyn-Cre was injected into the DG of *Htr1a*flox/flox mice. (**B**) Representative example showing photometry and EEG recordings in the DG of a control mouse injected with AAV9-CaMKII-GCaMP6s alone. Right, Fourier transformation of calcium activity during wake (blue) non-rapid eye movement (NREM) sleep (red). (**C**) A representative example showing

*Figure 6 continued on next page*

*Figure 6 continued*

photometry and EEG recordings in the DG of a mouse injected with AAV9-CaMKII-GCaMP6s and AAV1-hSyn-Cre. Right, Fourier transformation of calcium activity during wake (blue) NREM sleep (red). (**D**) Left, Quantification of the relative power of the calcium oscillation in the range of 1–2 cycles/min in the Cre and control groups (16 sessions from 5 mice for Cre, 16 sessions from 6 mice for control). Right, Quantification of calcium oscillation amplitudes in the Cre and control groups. Calcium signals in each mouse were normalized to Z scores. **p<0.01, ***p<0.001, unpaired t-test. (**E**) Schematic representation of the contextual fear conditioning (CFC) experimental design. (**F**) Left, Contextual fear recall tests showing percentage of freezing in 1 min time bins for *Htr1a*flox+/+ mice bilaterally injected with AAV9-CaMKII-Cre-GFP (Cre) or AAV9-CaMKII-GFP (GFP). Right, Quantification of freezing behavior over 5 min interval during contextual recall tests in Cre and GFP groups (N=11 for Cre, N=12 for GFP, *p<0.05, ***p<0.001, unpaired t-test).

The online version of this article includes the following figure supplement(s) for figure 6:

**Figure supplement 1.** Fluorescent in situ hybridization (FISH) of *Htr1a* receptors in the dentate gyrus (DG).

**Figure supplement 2.** Sleep and photometry analysis in *Htr1a* mice.

## Discussion

In this study, we demonstrated that major glutamatergic cell types of the DG, the GCs and MCs, display ISO activity (1–2 cycles/min or 0.02–0.03 Hz) during NREM sleep (*Figures 1 and 2*) but not during awake or REM periods. Our two-photon calcium imaging data shows that this oscillation is mainly driven by NREM-active glutamatergic cells (about 50% of GCs and 28% of MCs, *Figure 3*). Furthermore, we found that the ISO is negatively correlated with phasic 5-HT release in raphe nuclei during NREM sleep (*Figures 4 and 5*). Finally, we showed that local genetic ablation of *Htr1a* receptors in GCs impaired the ISO and fear memory recall a week after contextual fear conditioning (*Figure 6*). Taken together, our findings demonstrate that the serotonin system plays a pivotal role in modulating DG activity during NREM sleep.

MCs and GCs are two major cell populations of DG which are thought to be involved in pattern separation and other cognitive functions in awake animals (*Senzai and Buzsáki, 2017*; *GoodSmith et al., 2017*; *Danielson et al., 2017*; *Kim et al., 2020*). The activity of GCs and MCs has been investigated with both electrophysiological and imaging tools and the available data suggests that both cell types are significantly more active during sleep (*Senzai and Buzsáki, 2017*; *Pofahl et al., 2021*; *Jung and McNaughton, 1993*) than wakefulness. However, to our knowledge, ours is the first report showing that the population activity is dynamically rendered to ISO pattern during NREM sleep which is organized by at least one of the neuromodulatory systems, the 5-HT.

Two-site photometry recordings revealed a tight coupling between the activity of the raphe and the DG. During awake periods we detected a strong positive correlation between the two regions, whereas during NREM intervals this activity became strongly anticorrelated. In REM sleep, we detected a greater variability of correlation across animals, which cannot be attributed to serotonin, as most neuromodulatory systems are silent during REM sleep. Rather, varied DG activity during REM sleep across recording sessions may contribute to this variability.

There is much less available data on the activity pattern of MCs and GCs during MA episodes. Here, we showed that about 50% of putative GCs and 28% of putative MCs are upregulated during NREM and their activity is abruptly terminated by MAs.

We also observed that local downregulation of *Htr1a* receptors disrupts the ISO, suggesting that the 5-HT is directly involved in the regulatory mechanisms of this oscillatory process. *Htr1a* receptors are expressed by GCs which suggest that the inhibitory effect is mediated via this cell type. Since MCs are driven by GCs via highly efficient 'detonator' type of synapses, our current working hypothesis is that MC excitation is driven by GCs during NREM. We speculate that the drop of the calcium activity in MCs during the ISO is not mediated by serotonin but rather it is the result of the lack of excitation from GCs during the phasic 5-HT release. This hypothesis is supported by the difference in the delay preceding MAs in GCs and MCs (5.89+/-0.28 s in GCs *vs.* 4.89+/-0.40 s in MCs) which shows that GC inactivity is followed by MC inactivity. Simultaneous recording of MCs and GCs with two-photon calcium imaging in *Htr1a* knockout mice will further test this hypothesis.

ISOs have been observed in multiple brain regions during sleep-wake cycles (*Hughes et al., 2011*). Polysomnographic recordings have revealed a 0.02 Hz ISO pattern recorded from cortical areas (*Lecci et al., 2017*; *Vanhatalo et al., 2004*) which frequency is on the same time scale as the ISO we recorded from the DG. The cortical ISO is dominant in the sigma (10–15 Hz) power range in both mice

and humans (*Lecci et al., 2017*) which is the most prominent frequency band that contains neural rhythms associated with the gating of sensory information during sleep. A follow-up study by the same group demonstrated that the ISO from cortical EEG sigma band is correlated with MAs (*Cardis et al., 2021*). Our data with others (*Kato et al., 2022*) show that 5-HT concentration slowly oscillates in the hippocampus and in the raphe during NREM, thus it would be intriguing to conclude that peaks of the phasic 5-HT bursts act as the gating signal when the animal is more receptive to external stimulation and this can lead to awakening. However, we also need to point out that there are multiple other systems that can be active in parallel and contribute to the shaping of the sleep microarchitecture (*Kjaerby et al., 2022*). Several recent publications from different laboratories have shown rhythmic release of norepinephrine (NE) (~0.03 Hz) in the medial prefrontal cortex, the thalamus, and in the locus coeruleus (LC), and similar rhythmic activity of glutamatergic neurons in the preoptic area during sleep-wake cycles (*Kjaerby et al., 2022*; *Antila et al., 2022*; *Osorio-Forero et al., 2021*). The recording techniques used in these studies are highly similar to ours and the analyses arrive at the same conclusion, that is, about 30% of the phasic neuromodulator bursts lead to MAs while NREM sleep is maintained about 60% of the time. Furthermore, their work also demonstrated a correlation between LC-NE signal and EEG sigma power during NREM sleep, a characteristic EEG frequency band of sleep spindles. We performed a correlation analysis between DG activity and EEG sigma power and found a positive correlation between them during NREM sleep (*Figure 1—figure supplement 1C–E*). The available data obtained with other novel neuromodulatory sensors suggests, that several other neuromodulatory and peptidergic systems (i.e. histaminergic, cholinergic, oxytocinergic) (*Dong et al., 2023*; *Jing et al., 2020*; *Zhang et al., 2024*; *Smith et al., 2024*; *Osorio-Forero et al., 2025*) also display this phasic ISO activity during NREM sleep which raises the question of how the individual components of the ascending reticular activating system shapes sleep microarchitecture. Increase in the infraslow oscillatory frequency of NE during NREM leads to increased frequency of MAs (*Kjaerby et al., 2022*; *Antila et al., 2022*). However, increasing the extracellular 5-HT level by acute SSRI (selective serotonin reuptake inhibitors) administration does not seem to change the frequency of these events (*Kato et al., 2022*). This suggests that the 5-HT system might have different neuromodulatory functions from the NE system during sleep.

In our experiments we wanted to minimize the impact of the surgical procedures on the behavior, thus we used separate cohorts to record the photometry signals and carry out the behavior experiments, therefore, we are unable to correlate the magnitude changes in the serotonergic ISO and memory performance. However, in a recent paper published by Kjaerby et al. there seemed to be a direct correlation between the magnitude of the norepinephrine pulses and sleep behavior (*Kjaerby et al., 2022*). Gradual decrease of norepinephrine during NREM led to a fragmented sleep phenotype characterized by increased MA occurrence, decreased REM, and reduced spindle activity. The memory performance was also tested in a novel object recognition task and found to be diminished in the manipulation group. Serotonin has multiple roles in the brain, many of them show overlap with proposed functions of the noradrenergic system including regulation of plasticity, signaling reward or fearful stimuli (*Teixeira et al., 2018*; *Paquelet et al., 2022*). Therefore, we speculate that the modification of serotonin dynamics during sleep will most likely interfere with memory performance, however, further experiments are required to test this hypothesis.

Our data demonstrated that phasic release of 5-HT during sleep acts as an inhibitory input to the neuronal activity in DG via *Htr1a* receptors, thus it serves as a pacemaker signal. In addition, we speculate that there could be an excitatory input which is responsible for the elevated activity of GC and MC subpopulations during NREM. The DG is the main input node of the hippocampal tri-synaptic loop receiving glutamatergic inputs from distant brain areas such as the entorhinal cortex or the supramammillary nucleus (*Farrell et al., 2021*). Recent data have shown a similar ISO in the medial entorhinal cortex in head-fixed mice while moving on a rotating wheel (*Gonzalo Cogno et al., 2024*) which also transcended immobility epochs, but further data would be needed to confirm that they exist during NREM sleep as well.

A recent work from Pofahl et al. have used in vivo two-photon microscopy and showed that GCs display synchronous activity patterns during immobility *Pofahl et al., 2021* which might be related to the phenomenon we observed. However, the authors were not able to break down the immobility intervals to sleep stages due to the lack of simultaneous EEG/EMG recordings, nor did they assess the activity of the MCs in their work. Our results demonstrated the dynamic activity of DG cells during

sleep (*Figure 3*). Interestingly, we did not observe the ISO at the single-cell level under two-photon microscope. One possibility is that this was due to different sleep patterns between head-restricted and freely moving animals. We hypothesize that deep sleep is needed for the ISO. Even though we habituated our mice extensively to our two-photon setup to minimize stress, head-fixed posture may still result in differences in sleep behavior due to the slightly increased stress response caused by head-fixation (*Juczewski et al., 2020*). Another possibility is that the ISO mainly has a dendritic origin and manifests less at the somatic level. Future dendritic imaging in sleeping mice can test this possibility.

In summary, here we showed that calcium activity in the DG is highly increased during NREM sleep periods, and population activity is entrained to ISO. The activity of the serotonergic system is highly correlated with this oscillation, and rhythmic bursts of 5-HT maintain the oscillatory activity, via *Htr1a* receptors expressed by GCs. While our study highlights the role of neuromodulation in organizing neuronal activity during sleep, the direct relationship between these effects and sleep-dependent memory functions, such as memory consolidation, remains to be explicitly demonstrated.

# Methods

## Key resources table

| Reagent type (species) or resource | Designation | Source or reference | Identifiers | Additional information |
|---|---|---|---|---|
| Strain (C57BL/6 J) | C57BL/6 J mouse | JAX: #000664 | RRID:IMSR_JAX:000664 | |
| Strain (*Dock10*$^{Cre}$) | *Dock10*$^{Cre}$ mouse | MGI:6117432 | N/A | transgene insertion, by Susumu Tonegawa |
| Strain (*Drd2*$^{Cre}$) | *Drd2*$^{Cre}$ mouse | GENSAT | MMRRC:032108-UCD | |
| Strain (*Slc6a4*$^{Cre}$) | *Slc6a4*$^{Cre}$ mouse | JAX: #014554 | RRID:IMSR_JAX:014554 | Common name: SERT-Cre |
| Strain (*Htr1a*$^{flox/flox}$) | *Htr1a*$^{flox/flox}$ mouse | Rene Hen at Columbia University | N/A | Generated by Rene Hen |
| Strain (pAAV.Syn.Flex.GCaMP6s.WPRE.SV40) | AAV1-SYN-FLEX-GCaMP6s | Addgene | RRID:Addgene_100845 | |
| Strain (AAV.CamKII.GCaMP6s.WPRE.SV40) | AAV9-CamKIIa-GCaMP6s | Addgene | RRID:Addgene_107790 | |
| Strain (AAVdj-syn-jGCaMP7b) | AAVdj-syn-jGCaMP7b | Gene Vector and Virus Core at Stanford University | RRID:Addgene_104489 | Custom-made at Stanford virus core |
| Strain (AAV9-hSyn-GRAB$_{5-HT2h}$) | AAV9-hSyn-GRAB$_{5-HT2h}$ | Vigene Biosciences | Cat#YL10097-AV9 | |
| Strain (pENN.AAV.hSyn.Cre.WPRE.hGH) | AAV1-hSyn-Cre | Addgene | RRID:Addgene_105553 | |
| Strain (pENN.AAV.CamKII.HI.GFP-Cre.WPRE.SV40) | AAV9-CaMKII-Cre-GFP | Addgene | RRID:Addgene_105551 | |
| Strain (pENN.AAV.CamKII0.4.eGFP.WPRE.rBG) | AAV9-CaMKII-GFP | Addgene | RRID:Addgene_105541 | |
| Commercial assay (*Htr1a* RNAscope probe) | *Htr1a* mouse RNAscope probe | Advanced Cell Diagnostics | Cat#312301 | |
| Commercial assay (EGFP RNAscope probe) | EGFP RNAscope probe | Advanced Cell Diagnostics | Cat#400281 | |

## Animals

This study was carried out in accordance with the US National Institute of Health (NIH) guidelines for the care and use of laboratory animals and approved by the Animal Care and Use Committees of Columbia University (Protocol# AC-AABL9550) and New York State Psychiatric Institute (NYSPI-1661 and 1667). Adult (10–16 wk of age) were used for all experiments from both sexes. The following mouse lines were used in the current study: C57BL/6 J (JAX 000664), *Dock10*$^{Cre}$, *Drd2*$^{Cre}$, *Slc6a4*$^{Cre}$, *Htr1a*$^{flox/flox}$. Mice were housed in 12 hr light-dark cycles (lights on at 07:00 am and off at 07:00 pm). *Dock10*$^{Cre}$, *Drd2*$^{Cre}$, *Slc6a4*$^{Cre}$ were bred with C57BL/6 J mice, and the heterozygote offspring was used in the experiments. Homozygote offspring of the *Htr1a*$^{flox}$ line was used in local genetic manipulations.

## Surgical procedures

### EEG and Fiber implants

Mice were anesthetized with a mixture of ketamine and Xylazine (100 mg×kg-1 and 10 mg×kg-1, intraperitoneally), then placed on a stereotaxic frame with a closed-loop heating system to maintain body temperature. After asepsis, the skin was incised to expose the skull, and a small craniotomy (~0.5 mm in diameter) was made on the skull above the regions of interest. A solution containing 50–200 nl viral constructs was loaded into a pulled glass capillary and injected into the target region using a Nanoinjector (WPI). Optical fibers (0.2 mm diameter, 0.39 NA, Thorlabs) were implanted into the target region with the tip 0.1 mm above the virus injection site for fiber photometry recording. For EEG and EMG recordings, a reference screw was inserted into the skull on top of the cerebellum. EEG recordings were made from two screws on top of the cortex 1 mm from midline, 1.5 mm anterior to the bregma, and 1.5 mm posterior to the bregma, respectively. Two EMG electrodes were bilaterally inserted into the neck musculature. EEG screws and EMG electrodes were connected to a PCB board which was soldered with a 5-position pin connector. All the implants were secured onto the skull with dental cement (Lang Dental Manufacturing). After surgery, the animals were returned to their home cages for recovery for at least 2 wk before any experiment.

### Virus injection

For fiber photometry, 150–200 nl AAV1-FLEx-GCaMP6s or AAV9-hSyn-5-HT2h was unilaterally injected in the DG (AP –1.9 mm, ML 1.5 mm, DV 1.7 mm) or the dorsal raphe (AP –4.5 mm, ML 0 mm, DV 3.2 mm), respectively. An optical fiber was implanted 0.1 mm above the injection site. The DV Coordinates listed above are relative to the pial surface.

For two-photon imaging, the viral aliquots were diluted five times with physiological saline (titer: ~$5.36×10^{12}$ GC/ml) and 200 nl AAVdj-SYN-jGCaMP7b was injected unilaterally to the left DG (AP –1.9 mm, ML 1.5 mm, DV –1.7 mm).

### Optical cannula, EEG, and EMG implant for two-photon imaging

For two-photon imaging, we modified our standard surgical procedures (*Turi et al., 2019*) by implanting a pair of bone screws for unilateral EEG recording on the contralateral side to the cannula. Briefly, a week after the virus injection the mice were deeply anesthetized with Isoflurane, then a sagittal incision was made on the top of the skull. After removing the skin, trephination was done on the left side of the skull, by using a sterile 2 mm diameter tissue punch. The overlying cortical and hippocampal tissue was removed from above the DG, while the brain was constantly irrigated with sterile cortex buffer (125 mM NaCl, 5 mM KCl,10mM glucose, 10 mM HEPES, 2 mM CaCl2, 2 mM MgSO4. pH set to 7.4, osmolality to 305 mOsm). A sterile metal cannula (diameter: 2 mm, height: 1.8 mm) with a glass coverslip glued (Norland) to the bottom was implanted above the DG and secured in place with layers of tissue adhesive (3 M Vetbond) and dental acrylic. After the cannula was safely in place, we predrilled two holes on the right side of the skull (~1.5 mm from the middle line; 1.5 mm anterior and 1.5 mm posterior of the Bregma, respectively) for implanting recording electrodes, and one hole above the cerebellum for a reference electrode. Silver wires (125 μm diameter, PFA-coated) used for EEG recordings were placed into the predrilled holes and secured in place with stainless-steel skull screws (0.8 mm diam, 2.15 mm long). A pair of silver wires (125 μm diameter, PFA-coated) were implanted to the neck muscle as described in the fiber photometry section to facilitate EMG recordings. Finally, a metal head bar was secured to the skull with dental acrylic and the custom PCB interface with the connected silver wires and a connector was secured to the head bar with dental acrylic as well. All exposed bone areas were covered with dental acrylic and the mice were returned to their home cage for postoperative care and recovery.

## EEG recording

Mouse sleep behavior was monitored using EEG and EMG recording along with an infrared video camera at 30 frames per second. Recordings were performed for 24 hr (light on at 7:00 am and off at 7:00 pm) in a behavioral chamber inside a sound-attenuating cubicle (Med Associated Inc). Animals were habituated in the chamber for at least 4 hr before recording. EEG and EMG signals were recorded, bandpass filtered at 0.5–500 Hz, and digitized at 1017 Hz with 32-channel amplifiers (TDT,

PZ5, and RZ5D or Neuralynx Digital Lynx 4 S). For sleep analysis, spectral analysis was carried out using a fast Fourier transform (FFT) over a 5 s sliding window, sequentially shifted by 2 s increments (bins). Brain states were semi-automatically classified into wake, NREM sleep, and REM sleep states using a custom-written MATLAB program using the following scoring criteria: wake: desynchronized EEG and high EMG activity; NREM: synchronized EEG with high-amplitude, delta frequency (0.5–4 Hz) activity and low EMG activity; REM: high power at theta frequencies (6–9 Hz) and low EMG activity. Semi-auto classification was validated manually by trained experimenters. We categorized wake bouts of <15 s as microarousals (MA) in the following way: the baseline of the EMG was calculated by averaging the signal, then a threshold of 0.5 standard derivation above the baseline was used to detect the MA. Finally, each MA event was further validated manually by a trained experimenter.

## Two-photon recording

Mice were extensively habituated to the two-photon rig and to the head-fixed position before the recording sessions. They were placed in the head holder apparatus for a gradually increasing amount of time (from 5 min to 1 hr) over the course of 2 wk. Mice were head-restrained under the objective of the two-photon microscope, but otherwise able to walk on a custom-built treadmill. The mouse's behavior on the treadmill was collected with an earlier version of BehaviorMate (*Bowler, 2023*). The locomotory and stationary epochs were detected with a rotary encoder attached to the axel of the treadmill wheel. The signal from the encoder was processed with BehaviorMate.

A commercial in vivo multiphoton imaging system was used for the two-photon recordings (Bruker Ultima IV), equipped with a dual scan head. Nikon 10 x (0.4 NA) or 16 x (0.8 NA) objectives were used to record cellular activity. The emitted light was collected with photomultiplier tubes (Hamamatsu, H11706-40 GaAsP). Excitation was achieved with an InSight X3 (Spectra-Physics) laser tuned to 940 nm. The light intensity was regulated with pockels cells (Model 350–80-LA-02 KD*P Series E-O Modulator, Conoptics). The collection optics consisted of a standard filter set (565 dichroic, 525/70, and 595/50 bandpass) before the detectors.

Two-photon imaging sessions took 45–60 min. Images were collected at a 10 Hz frequency sampling rate in resonant scanning mode. Image dimensions were set to 512×512 pixels (width and height). The collected binary data was directly converted to Hierarchical Data Format (HDF) then it was deposited to our server. Image registration and cell detection was performed using Suite2p software (ver. 0.7.0) package (*Pachitariu et al., 2017*). The registered and segmented images went through hand curation by an experienced investigator. During this process, the ROI selection in the GC layer was restricted to cells with small soma size and sparse activity patterns while ambiguous ROIs (non-cellular structures, cells with big soma size and high activity) were discarded from the segmented data. In the hilar recordings only large, multipolar cells were accepted as putative MCs which had large amplitude calcium transients with fast onset and exponential decay. Delta F/F was calculated using the extracted raw calcium traces and neuropil signals with custom-written code in Python as to the following. The unfiltered neuropil was first subtracted from the unfiltered raw trace. Two partial baselines were estimated then by filtering the residual raw trace, and the neuropil. The residual raw trace minus its partial baseline forms the numerator term of the dF/F. The denominator is the sum of the partial baselines. The partial baselines were calculated by applying a minimax filter to the respective (smoothed) traces.

To detect wake/sleep states, a Neuralynx Digital Lynx 4 S system was used for EEG/EMG recordings under the two-photon microscopy. EMG and EEG signals were acquired and filtered as described above. The EEG/EMG data was collected on a PC. Sleep states were calculated as described above. The up-regulated (i.e. more active during NREM sleep), down-regulated (i.e. less active during NREM sleep), and non-significant cells were defined by comparing each cell's activity between epochs of quiet wake (without motion) and NREM sleep (unpaired t-test).

To facilitate sleep under head-fixed conditions, the mice underwent mild sleep deprivation during the night (from 5 pm to 8 am) before the recording session. Please note that the night is the active phase for mice. The mice were placed in a custom-designed cage within a sound attenuated enclosure on a treadmill controlled by a microcontroller. The treadmill was set to 5 s on and 10 s off cycles. On the morning of the two-photon recording sessions, the mice were transferred to the head-fixed apparatus from the sleep deprivation box.

## Fiber photometry

Fiber photometry recordings were performed as previously described (*Teng et al., 2022*). In brief, calcium dependent GCaMP fluorescence was excited by sinusoidal modulated LED light (473 nm, 220 Hz; 405 nm, 350 Hz, Doric lenses) and detected by a femtowatt silicon photoreceiver (New Port, 2151). Photometric signals and EEG/EMG signals were simultaneously acquired by a real-time processor (RZ5D, TDT) and synchronized with behavioral video recording. A motorized commutator (ACO32, TDT) was used to route electric wires and optical fiber. The collected data were analyzed by custom MATLAB scripts. They were first extracted and subject to a low-pass filter at 2 Hz. A least-squares linear fit was then applied to produce a fitted 405 nm signal. The DF/F was calculated as: (F-F0)/F0, where F0 was the fitted 405 nm signals. To compare activity across animals, photometric data were further normalized using Z-score calculation in each mouse. To analyze the infraslow oscillation (ISO) of calcium signals, data were first downsampled to 1 Hz, then spectral analysis was carried out using FFT over a 2 min sliding window, sequentially shifted by 2 s increments (bins). The spectral power in the range of 0–6 cycles per min (i.e. 0–0.1 Hz) during wake and sleep was analyzed. Then, the power in the range of 1–2 cycles per min (i.e. 0.0167–0.033 Hz) was normalized to the total power and used for statistical comparison between *Htr1a*$^{flox/flox}$ and control mice in *Figure 6D*. To detect the calcium trough of ISO in *Figures 1 and 2*, we first calculated a moving baseline by smoothing the calcium signals over 60 s, then set a threshold (0.2 standard deviation from the moving baseline) for events of calcium decrease, and finally detected the minimum point in each event as the calcium trough. The similar method was used to detect 5-HT peaks in *Figure 4*, *Figure 4—figure supplement 1*. To examine the infraslow oscillation at different stages of NREM sleep, we calculated the oscillation power and amplitude at the first (T0), early (T1), and late (T2) 1 min of NREM sleep epochs in *Figure 1—figure supplement 1B*. T0 refers to the first minute of the first NREM epochs, defined as those with the prior wakefulness longer than 5 min. T1 refers to the first minute of all NREM epochs, whereas T2 refers to the last minute of all NREM epochs of the recording sessions. To analyze the relationship between DG and raphe activity in *Figure 5*, the Pearson correlation coefficient (r) was calculated between two signals during wakefulness, NREM sleep, and REM sleep in each recording session.

To examine the DG activity during microarousals in *Figures 1 and 2* and *Figure 6—figure supplement 2*, the calcium signals were aligned to the onset of each microarousal event to generate the per-stimulus time histogram (PSTH). We then defined the baseline by calculating the average of calcium signals in the first 10 s window before the MA (from –15 s to –5 s). The onset of calcium decline is defined as the timepoint where calcium decrease was larger than 0.05 standard deviation from this baseline. Finally, the time difference between the onset of calcium declines and the MA onset was named as latency. The difference between the baseline and the calcium trough was named as calcium drop. The same method was used to calculate the latency of 5-HT release during MA in *Figure 4—figure supplement 1*, where calcium decrease was replaced with 5-HT increase.

## Fluorescence in situ hybridization (FISH)

Mice were deeply anesthetized and then decapitated and their brains were snap-frozen on powdered dry ice. The fresh frozen brains were sectioned at 20 µm thickness using a cryostat. FISH was performed using RNAscope Multiplex Fluorescent Assay V2 (Advanced Cell Diagnostics) as per the manufacturer's recommendations. Reagents: Htr1a in situ hybridization probe: cat# 312301, GFP in situ hybridization probe: cat# 400281 (Advanced Cell Diagnostics). Images were acquired using a Zeiss 810 confocal microscope.

## Histology

Viral expression and placement of optical implants were verified at the termination of the experiments using DAPI counterstaining of 100 µm coronal sections (Prolong Gold Antifade Mountant with DAPI, Invitrogen). Images were acquired using a Zeiss 810 confocal microscope. Cell numbers were counted manually in ImageJ.

## Statistics

No method of randomization was used to determine how animals were allocated to experimental groups. Investigators were not blinded to group allocations. Mice in which the *post hoc* histological examination showed off-target viral injection or fiber implantation were excluded from further

analysis. Paired and unpaired t-tests were used and are indicated in the respective figure legends. All analyses were performed in MATLAB. Data are presented as mean ± s.e.m.

## Code availability

MATLAB scripts for EEG and photometry analysis are available on GitHub (https://github.com/thepen-glab/TDTEEG, copy archived at *Peng, 2025*).

## Acknowledgements

We thank René Hen at Columbia University for providing Htr1a$^{flox/flox}$ mice. We thank Kitti Rusznak and Hayley Judice for helping with behavioral experiments. The 5-HT sensor (AAV9-hSyn-5-HT2h) was a gift from the Yulong Li's laboratory at Peking University. This work was supported by startup funds from Columbia University, Columbia University Precision Medicine Initiative, and NINDS R01NS12997 to YP. GFT was supported by a BBRF Young Investigator Award, a NIMH R21MH122965 and NIA R21AG079025. The AAVdj-SYN-jGCaMP7b virus was packaged by the Gene Vector and Virus Core at Wu Tsai Neurosciences Institute at Stanford University.

## Additional information

### Funding

| Funder | Grant reference number | Author |
|---|---|---|
| Columbia University | Startup funds | Yueqing Peng |
| National Institute of Mental Health | R21MH122965 | Gergely F Turi |
| National Institute on Aging | R21AG079025 | Gergely F Turi |
| National Institute of Neurological Disorders and Stroke | R01NS129997 | Yueqing Peng |
| Brain & Behavior Research Foundation | Young Investigator Award | Gergely F Turi |

The funders had no role in study design, data collection and interpretation, or the decision to submit the work for publication.

### Author contributions

Gergely F Turi, Conceptualization, Resources, Data curation, Formal analysis, Supervision, Funding acquisition, Investigation, Methodology, Writing – original draft, Writing – review and editing; Sasa Teng, Data curation, Formal analysis, Investigation, Methodology, Writing – original draft, Writing – review and editing; Xinyue Chen, Emily CY Lim, Carla Dias, Ruining Hu, Ruizhi Wang, Fenghua Zhen, Data curation, Investigation, Methodology; Yueqing Peng, Conceptualization, Resources, Data curation, Software, Formal analysis, Supervision, Funding acquisition, Validation, Investigation, Visualization, Methodology, Writing – original draft, Project administration, Writing – review and editing

### Author ORCIDs

Gergely F Turi ● https://orcid.org/0000-0001-5651-9459
Xinyue Chen ● https://orcid.org/0000-0003-0264-1488
Yueqing Peng ● https://orcid.org/0000-0002-3488-0670

### Ethics

This study was carried out in accordance with the US National Institute of Health (NIH) guidelines for the care and use of laboratory animals and approved by the Animal Care and Use Committees of Columbia University and New York State Psychiatric Institute. (Protocol# AC-AABL9550).

Reviewer #1 (Public review): https://doi.org/10.7554/eLife.100196.4.sa1
Reviewer #2 (Public review): https://doi.org/10.7554/eLife.100196.4.sa2

Author response https://doi.org/10.7554/eLife.100196.4.sa3

## Additional files

### Supplementary files
MDAR checklist

Source data 1. Source data for the main figures and figure supplements.

### Data availability
Source data supporting the findings of this study are provided in *Source data 1*.

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
