## [Editor Report · eLife Assessment]

This **important** study shows that a very slow (infraslow) oscillation occurs in calcium recordings from the dentate gyrus of the adult mouse. The authors suggest that it is related to sleep stage and serotonin acting at one type of serotonin receptor in the dentate gyrus. The results are significant because they suggest new insight into how a slow oscillation affects memory through serotonin receptors in the dentate gyrus. **Convincing** data are provided to support the claims.

---

## [Referee Report · Reviewer #1 (Public review)]

Turi, Teng and the team used state of the art techniques to provide convincing evidence on the infraslow oscillation of DG cells during NREM sleep, and how serotonergic innervation modulates hippocampal activity pattern during sleep and memory. First, they showed that the glutamatergic DG cells become activated following an infraslow rhythm during NREM sleep. In addition, the infraslow oscillation in the DG is correlated with rhythmic serotonin release during sleep. Finally, they found that specific knockdown of 5-HT receptors in the DG impairs the infraslow rhythm and memory, suggesting that serotonergic signaling is crucial for regulating DG activity during sleep. Given that the functional role of infraslow rhythm still remains to be studied, their findings deepen our understanding on the role of DG cells and serotonergic signaling in regulating infraslow rhythm, sleep microarchitecture and memory.

---

## [Referee Report · Reviewer #2 (Public review)]

Summary:

The authors investigated DG neuronal activity at the population and single cell level across sleep/wake periods. They found an infraslow oscillation (0.01-0.03 Hz) in both granule cells (GC) and mossy cells (MC) during NREM sleep. The important findings are:

(1) The antiparallel temporal dynamics of DG neuron activities and serotonin neuron activities/extracellular serotonin levels during NREM sleep

(2) The GC Htr1a-mediated GC infraslow oscillation.

Strengths:

(1) The combination of polysomnography, Ca-fiber photometry, two-photon microscopy and gene depletion is technically sound. The coincidence of microarousals and dips in DG population activity is convincing. The dip in activity in upregulated cells is responsible for the dip at the population level.

(2) DG GCs express excitatory Htr4 and Htr7 in addition to inhibitory Htr1a, but deletion of Htr1a is sufficient to disrupt DG GC infraslow oscillation, supporting the importance of Htr1a in DG activity during NREM sleep.

Weaknesses from the original round of review:

(1) The current data set and analysis are insufficient to interpret the observation correctly [...].

(2) It is acceptable that DG Htr1a KO induces the reduced freezing in the CFC test (Fig. 6E, F), but it is too much of a stretch that the disruption of DG ISO causes impaired fear memory. There should be a correlation.

(3) It is necessary to describe the extent of AAV-Cre infection. The authors injected AAV into the dorsal DG (AP -1.9 mm), but the histology shows the ventral DG (Supplementary Fig. 4), which reduces the reliability of this study.

Comments on revisions:

Thank you for the clarification of the detection criteria and the quantification of the specific events. This reviewer can now follow the authors' interpretation.

---

## [Author Response]

The following is the authors’ response to the previous reviews.

**Public Reviews:**

**Reviewer #1 (Public review):**
Turi, Teng and the team used state-of-the-art techniques to provide convincing evidence on the infraslow oscillation of DG cells during NREM sleep, and how serotonergic innervation modulates hippocampal activity pattern during sleep and memory. First, they showed that the glutamatergic DG cells become activated following an infraslow rhythm during NREM sleep. In addition, the infraslow oscillation in the DG is correlated with rhythmic serotonin release during sleep. Finally, they found that specific knockdown of 5-HT receptors in the DG impairs the infraslow rhythm and memory, suggesting that serotonergic signaling is crucial for regulating DG activity during sleep. Given that the functional role of infraslow rhythm still remains to be studied, their findings deepen our understanding on the role of DG cells and serotonergic signaling in regulating infraslow rhythm, sleep microarchitecture and memory.
**Reviewer #2 (Public review):**
Summary:The authors investigated DG neuronal activity at the population and single cell level across sleep/wake periods. They found an infraslow oscillation (0.01-0.03 Hz) in both granule cells (GC) and mossy cells (MC) during NREM sleep. The important findings are (1) the antiparallel temporal dynamics of DG neuron activities and serotonin neuron activities/extracellular serotonin levels during NREM sleep, and (2) the GC Htr1a-mediated GC infraslow oscillation.Strengths:(1) The combination of polysomnography, Ca-fiber photometry, two-photon microscopy and gene depletion is technically sound. The coincidence of microarousals and dips in DG population activity is convincing. The dip in activity in upregulated cells is responsible for the dip at the population level.(2) DG GCs express excitatory Htr4 and Htr7 in addition to inhibitory Htr1a, but deletion of Htr1a is sufficient to disrupt DG GC infraslow oscillation, supporting the importance of Htr1a in DG activity during NREM sleep.Weaknesses:(1) The current data set and analysis are insufficient to interpret the observation correctly.a. In Fig 1A, during NREM, the peaks and troughs of GC population activities seem to gradually decrease over time. Please address this point.b. In Fig 1F, about 30% of Ca dips coincided with MA (EMG increase) and 60% of Ca dips did not coincide with EMG increase. If this is true, the readers can find 8 Ca dips which are not associated with MAs from Fig 1E. If MAs were clustered, please describe this properly.c. In Fig 1F, the legend stated the percentage during NREM. If the authors want to include the percentage of wake and REM, please show the traces with Ca dips during wake and REM. This concern applies to all pie charts provided by the authors.d. In Fig 1C, please provide line plots connecting the same session. This request applies to all related figures.e. In Fig 2C, the significant increase during REM and the same level during NREM are not convincing. In Fig 2A, the several EMG increasing bouts do not appear to be MA, but rather wakefulness, because the duration of the EMG increase is greater than 15 seconds. Therefore, it is possible that the wake bouts were mixed with NREM bouts, leading to the decrease of Ca activity during NREM. In fact, In Fig 2E, the 4th MA bout seems to be the wake bout because the EMG increase lasts more than 15 seconds.f. Fig 5D REM data are interesting because the DRN activity is stably silenced during REM. The varied correlation means the varied DG activity during REM. The authors need to address it.g. In Fig 6, the authors should show the impact of DG Htr1a knockdown on sleep/wake structure including the frequency of MAs. I agree with the impact of Htr1a on DG ISO, but possible changes in sleep bout may induce the DG ISO disturbance.(2) It is acceptable that DG Htr1a KO induces the reduced freezing in the CFC test (Fig. 6E, F), but it is too much of a stretch that the disruption of DG ISO causes impaired fear memory. There should be a correlation.(3) It is necessary to describe the extent of AAV-Cre infection. The authors injected AAV into the dorsal DG (AP -1.9 mm), but the histology shows the ventral DG (Supplementary Fig. 4), which reduces the reliability of this study.

Responses to weaknesses mentioned above have been addressed in the first revision.

Comments on revisions:In the first revision, I pointed out the inappropriate analysis of the EEG/EMG/photometry data and gave examples. The authors responded only to the points raised and did not seem to see the need to improve the overall analysis and description. In this second revision, I would like to ask the authors to improve them. The biggest problem is that the detection criteria and the quantification of the specific event are not described at all in Methods and it is extremely difficult to follow the statement. All interpretations are made by the inappropriate data analysis; therefore, I have to say that the statement is not supported by the data.Please read my following concerns carefully and improve them.(1) The definition of the event is critical to the detection of the event and the subsequent analysis. In particular, the authors explicitly describe the definition of MA (microarousal), the trough and peak of the population level of intracellular Ca concentrations, or the onset of the decline and surge of Ca levels.(1-1) The authors categorized wake bouts of <15 seconds with high EMG activity as MA (in Methods). What degree of high EMG is relevant to MA and what is the lower limit of high EMG? In Fig 1E, there are some EMG spikes, but it was unclear which spike/wave (amplitude/duration) was detected as MA-relevant spike and which spike was not detected. In Fig 2E, the 3rd MA coincides with the EMG spike, but other EMG spikes have comparable amplitude to the 3rd MA-relevant EMG spike. Correct counting of MA events is critical in Fig 1F, 2F, 4C.

We have added more information about the MA definition in Methods, including EMG amplitude. Furthermore, we have re-analyzed MA and MA-related calcium signals in Fig1 and Fig2. Fig-S1 shows the traces of EMG aptitude for all MA events show in Fig1G and Fig2G.

(1-2) Please describe the definition of Ca trough in your experiments. In Fig 1G, the averaged trough time is clear (~2.5 s), so I can acknowledge that MA is followed by Ca trough. However, the authors state on page 4 that "30% of the calcium troughs during NREM sleep were followed by an MA epoch". This discrepancy should be corrected.

We apologize for the misleading statement. We meant 30% of ISO events during NERM sleep. We have corrected this. To detect the calcium trough of ISO, we first calculated a moving baseline (blue line in Fig-S2 below) by smoothing the calcium signals over 60 s, then set a threshold (0.2 standard deviation from the moving baseline) for events of calcium decrease, and finally detected the minimum point (red dots in Fig-S2) in each event as the calcium trough. We have added these in Methods.

(1-3) Relating comment 1-2, I agree that the latency is between MA and Ca through in page 4, as the authors explain in the methods, but, in Fig 1G, t (latency) is labeled at incorrect position. Please correct this.

We are sorry for the mistake in describing the latency in the Methods. The latency was defined as the time difference between the onset of calcium decline (see details below in 1-4) and the onset of the MA. We have corrected this in the revised manuscript. Thus, the labeling in Fig1G was correct.

(1-4) The authors may want to determine the onset of the decline in population Ca activity and the latency between onset and trough (Fig 1G, latency t). If so, please describe how the onset of the decline is determined. In Fig 1G, 2G, S6, I can find the horizontal dashed line and infer that the intersection of the horizontal line and the Ca curve is considered the onset. However, I have to say that the placement of this horizontal line is super arbitrary. The results (t and Drop) are highly dependent on the position of horizontal line, so the authors need to describe how to set the horizontal line.

Indeed, we used the onset of calcium decline to calculate the latency as mentioned above. First, we defined the baseline (dashed line in Fig1G) by calculating the average of calcium signals in the10s window before the MA (from -15s to -5s in Fig1G). The onset of calcium decline is defined as the timepoint where calcium decrease was larger than 0.05 SD from this baseline. We have added these in Methods.

(1-5) In order to follow Fig 1F correctly, the authors need to indicate the detection criteria of "Ca dip (in legend)". Please indicate "each Ca dip" in Fig 1E. As a reader, I would like to agree with the Ca dip detection of this Ca curve based on the criteria. Please also indicate "each Ca dip" in Fig 2E and 2F. In the case of the 2nd and 3rd MAs, do they follow a single Ca dip or does each MA follow each Ca dip? This chart is highly dependent on the detection criteria of Ca dip.

We have indicated each ca dip in Fig 1 and Fig 2.

As I mentioned above, most of the quantifications are not based on the clear detection criteria. The authors need to re-analyze the data and fix the quantification. Please interpret data and discuss the cellular mechanism of ISO based on the re-analyzed quantification.

As suggested, we have re-analyzed the MA and MA-related photometry signals. Accordingly, parts of Fig1 and Fig2 have been revised. Although there are some small changes, the main results and conclusions remain unchanged.

**Reviewer #3 (Public review):**
Summary:The authors employ a series of well-conceived and well-executed experiments involving photometric imaging of the dentate gyrus and raphe nucleus, as well as cell-type specific genetic manipulations of serotonergic receptors that together serve to directly implicate serotonergic regulation of dentate gyrus (DG) granule (GC) and mossy cell (MC) activity in association with an infra slow oscillation (ISO) of neural activity has been previously linked to general cortical regulation during NREM sleep and microarousals.Strengths:There are a number of novel and important results, including the modulation of dentage granule cell activity by the infraslow oscillation during NREM sleep, the selective association of different subpopulations of granule cells to microarousals (MA), the anticorrelation of raphe activity with infraslow dentate activity.The discussion includes a general survey of ISOs and recent work relating to their expression in other brain areas and other potential neuromodulatory system involvement, as well as possible connections with infraslow oscillations, micro arousals, and sensory sensitivity.Weaknesses:- The behavioral results showing contextual memory impairment resulting from 5-HT1a knockdown are fine, but are over-interpreted. The term memory consolidation is used several times, as well as references to sleep-dependence. This is not what was tested. The receptor was knocked down, and then 2 weeks later animals were found to have fear conditioning deficits. They can certainly describe this result as indicating a connection between 5-HT1a receptor function and memory performance, but the connection to sleep and consolidation would just be speculation. The fact that 5-HT1a knockdown also impacted DG ISOs does not establish dependency. Some examples of this are:– The final conclusion asserts "Together, our study highlights the role of neuromodulation in organizing neuronal activity during sleep and sleep-dependent brain functions, such as memory.", but the reported memory effects (impairment of fear conditioning) were not shown to be explicitly sleep-dependent.– Earlier in the discussion it mentions "Finally, we showed that local genetic ablation of 5-HT1a receptors in GCs impaired the ISO and memory consolidation". The effect shown was on general memory performance - consolidation was not specifically implicated.– The assertion on page 9 that the results demonstrate "that the 5-HT is directly acting in the DG to gate the oscillations" is a bit strong given the magnitude of effect shown in Fig. 6D, and the absence of demonstration of negative effect on cortical areas that also show ISO activity and could impact DG activity (see requested cortical sigma power analysis).– Recent work has shown that abnormal DG GC activity can result from the use of the specific Ca indicator being used (GCaMP6s). (Teng, S., Wang, W., Wen, J.J.J. et al. Expression of GCaMP6s in the dentate gyrus induces tonic-clonic seizures. Sci Rep 14, 8104 (2024). https://doi.org/10.1038/s41598-024-58819-9). The authors of that study found that the effect seemed to be specific to GCaMP6s and that GCaMP6f did not lead to abnormal excitability. Note this is of particular concern given similar infraslow variation of cortical excitability in epilepsy (cf Vanhatalo et al. PNAS 2004). While I don't think that the experiments need to be repeated with a different indicator to address this concern, you should be able to use the 2p GCaMP7 experiments that have already been done to provide additional validation by repeating the analyses done for the GCaMP6s photometry experiments. This should be done anyway to allow appropriate comparison of the 2p and photometry results.– While the discussion mentions previous work that has linked ISOs during sleep with regulation of cortical oscillations in the sigma band, oddly no such analysis is performed in the current work even though it is presumably available and would be highly relevant to the interpretation of a number of primary results including the relationship between the ISOs and MAs observed in the DG and similar results reported in other areas, as well as the selective impact of DG 5-HT1a knockdown on DG ISOs. For example, in the initial results describing the cross correlation of calcium activity and EMG/EEG with MA episodes (paragraph 1, page 4), similar results relating brief arousals to the infraslow fluctuation in sleep spindles (sigma band) have been reported also at .02 Hz associated with variation in sensory arousability (cf. Cardis et al., "Cortico-autonomic local arousals and heightened somatosensory arousability during NREMS of mice in neuropathic pain", eLife 2021). It would be important to know whether the current results show similar cortical sigma band correlations. Also, in the results on ISO attenuation following 5-HT1 knockdown on page 7 (fig. 6), how is cortical EEG affected? is ISO still seen in EEG but attenuated in DG?– The illustrations of the effect of 5-HT1a knockdown shown in Figure 6 are somewhat misleading. The examples in panels B and C show an effect that is much more dramatic than the overall effect shown in panel D. Panels B and C do not appear to be representative examples. Which of the sample points in panel D are illustrated in panels B, C? it is not appropriate to arbitrarily select two points from different animals for comparison, or worse, to take points from the extremes of the distributions. If the intent is to illustrate what the effect shown in D looks like in the raw data, then you need to select examples that reflect the means shown in panel D. It is also important to show the effect on cortical EEG, particularly in sigma band to see if the effects are restricted to the DG ISOs. It would also be helpful to show that MAs and their correlations as shown in Fig 1 or G as well as broader sleep architecture are not affected.– On page 9 of the results it states that GCs and MCs are upregulated during NREM and their activity is abruptly terminated by MAs through a 5-HT mediated mechanism. I didn't see anything showing the 5-HT dependence of the MA activity correlation. The results indicate a reduction in ISO modulation of GC activity but not the MA correlated activity. I would like to see the equivalent of Fig 1,2 G panels with the 5-HT1a manipulation.

Responses to Revewer#3 have been addressed in the first revision.

**Reviewer #1 (Recommendations for the authors):**
Minor comment: Several recent publications from different laboratories have shown rhythmic release of norepinephrine (NE) (~0.03 Hz) in the medial prefrontal cortex, the thalamus, and in the locus coeruleus (LC) of the mouse during sleep-wake cycles-> Please add "preoptic area" here

We have added the citation.

**Reviewer #2 (Recommendations for the authors):**
Minor(1) (abstract, page 2 line 9) what kind of "increased activity" did the authors find?

Increased activity compared to that during wakefulness. We have added this.

(2) (result, page 4) please define first, early, and late stage of NREM sleep in the methods.

We have added these in the Methods.

(3) (result, page 6) please define "the risetime of the phasic increase".

It refers to the latency between the increase of 5-HT and the MA onset. We have clarified this in the text.

(4) (supplement Fig 3 legend) please reword "5-HT events" and "5-HT signals" because these are ambiguous.

We have defined the events in the legend.

(5) (Fig 5A) please replace the picture without bubbles.

We have replaced the image in Fig5A.